# Electron Donor–Acceptor Capacity of Selected Pharmaceuticals against COVID-19

**DOI:** 10.3390/antiox10060979

**Published:** 2021-06-18

**Authors:** Ana Martínez

**Affiliations:** Departamento de Materiales de Baja Dimensionalidad, Instituto de Investigaciones en Materiales, Universidad Nacional Autónoma de México, Circuito Exterior SN, Ciudad Universitaria, Ciudad de México CP 04510, Mexico; martina@unam.mx

**Keywords:** COVID-19, SARS-CoV2, coronavirus, oxidation, electroaccepting power, DAM

## Abstract

More than a year ago, the first case of infection by a new coronavirus was identified, which subsequently produced a pandemic causing human deaths throughout the world. Much research has been published on this virus, and discoveries indicate that oxidative stress contributes to the possibility of getting sick from the new SARS-CoV-2. It follows that free radical scavengers may be useful for the treatment of coronavirus 19 disease (COVID-19). This report investigates the antioxidant properties of nine antivirals, two anticancer molecules, one antibiotic, one antioxidant found in orange juice (Hesperidin), one anthelmintic and one antiparasitic (Ivermectin). A molecule that is apt for scavenging free radicals can be either an electron donor or electron acceptor. The results I present here show Valrubicin as the best electron acceptor (an anticancer drug with three F atoms in its structure) and elbasvir as the best electron donor (antiviral for chronic hepatitis C). Most antiviral drugs are good electron donors, meaning that they are molecules capable of reduzing other molecules. Ivermectin and Molnupiravir are two powerful COVID-19 drugs that are not good electron acceptors, and the fact that they are not as effective oxidants as other molecules may be an advantage. Electron acceptor molecules oxidize other molecules and affect the conditions necessary for viral infection, such as the replication and spread of the virus, but they may also oxidize molecules that are essential for life. This means that the *weapons* used to defend us from COVID-19 may also harm us. This study posits the idea that oxide reduction balance may help explain the toxicity or efficacy of these drugs. These results represent a further advance on the road towards understanding the action mechanisms of drugs used as possible treatments for COVID-19. Looking ahead, clinical studies are needed to define the importance of antioxidants in treating COVID-19.

## 1. Introduction

As we all know and are consequently suffering, the novel coronavirus disease-19 (COVID-19) has produced a global pandemic [1]. To control the pandemic, governments have requested the population to maintain a mutual healthy distance and stay at home. This has evident negative effects on the economy. In Mexico, social inequality is particularly shameful. Fifty percent of the population lives in extreme poverty, surviving on daily earnings. For these people, it is counterproductive to stay at home and; therefore, implementing social distancing and obligatory lockdown, as the only means to control the pandemic, has not been readily accepted. Effective vaccines to prevent infection have already been developed, also in patients with chronic illness as autoimmune diseases [2], but unfortunately we do not have efficacious pharmaceutical *weapons* against the COVID-19 disease.

The capacity of various drugs to target specific weaknesses of the coronavirus has been assayed [3], but an effective and definitive antiviral strategy has yet to be formulated. The data published in the literature on the effects of vitamin D supplementation are still controversial in patients with COVID-19, but the importance of vitamin D in promoting the immune response to infections such as Covid19 has been reported [4]. The urgency of the situation we are experiencing demands and welcomes drug treatment strategies, and several are already on the market or undergoing clinical trials [5,6,7,8,9,10,11,12,13,14,15,16,17,18,19,20,21,22,23,24,25,26]. Promising drugs have been suggested as potential inhibitors of the main SARS-Cov2 protease, including those reported in Table 1. Apparently, some of the drugs are incorporated into the genome of RNA viruses, producing mutations and acting in the induction of errors in the replication of the virus, an outcome known as a viral error catastrophe. As Table 1 indicates, remedies used until now to control COVID-19 include antibiotics, antivirals, anticancer, antiretroviral, antiparasitic, anthelminthic, antineuroinflammatory and anti-neurodegenerative drugs.

Several studies previously suggested that the balance of disulfide-thiol is important for COVID-19 viral infection, and oxidative stress from free radicals can affect this balance. It has been reported [14,22,24] that oxidative stress contributes to increasing people’s vulnerability to different viruses. Although we cannot avoid oxidative stress in normal life, it is apparent that quarantine has negative effects on lifestyle and increases oxidative stress among the population. Likewise, indications that the severity of COVID-19 disease often relates to people’s age are possibly explained by the high antioxidant capacity of children compared to older adults. Therefore, free radical scavengers may be useful for curtailing the gravity of this disease and, for this reason, it is essential to investigate the antioxidant capacity of any drugs used for treating COVID-19. 

Although there are already a number of studies, no theoretical investigations have assessed the free radical scavenger capacity of these drugs. Thus, the main objective of this report is to evaluate the antioxidant properties of the molecules presented in Table 1. The single electron transfer mechanism is analyzed and a classification concerning the free radical scavenger capacity is provided. This information may help to identify the best *weapons* against COVID-19.

## 2. Computational Details

Gaussian09 was used for all electronic calculations [27]. Geometry optimizations without symmetry constraints were implemented at the M062x/6-311+G(2d,p) level of theory [28,29,30,31,32]. Harmonic analyses were calculated to verify local minima (zero imaginary frequencies). PubChem configurations were used as initial geometries.

Conceptual density functional theory (CDFT) is a chemical reactivity theory founded on density functional theory -based concepts [33,34,35,36,37,38,39]. Within this theory, there are response functions that help us to understand the chemical reactivity. The response functions that we used in this investigation are the electro-donating (ω−) and electro-accepting (ω+) powers, previously reported by Gázquez et al. [38,39]. These authors defined the propensity to donate charge or ω− as follows:ω− = (3I + A)2/16 (I − A)(1)
whereas the propensity to accept charge or ω+ is defined as
ω+ = (I + 3A)2/16 (I − A)(2)

I and A are vertical ionization energy and vertical electron affinity, respectively. Lower values of ω− imply greater capacity for donating charge. Higher values of ω+ imply greater capacity for accepting charge. In contrast to I and A, ω− and ω+ refer to charge transfers, not necessarily from one electron. This definition is based on a simple charge transfer model expressed in terms of chemical potential and hardness. These chemical descriptors have been used successfully in many different chemical systems [40,41,42,43,44,45,46,47]. 

## 3. Results and Discussion

Figure 1 reports the schematic representation of the studied molecules. Nine antivirals, two anticancer molecules, one antibiotic, one antioxidant that is found in orange juice (Hesperidin), one anthelminthic and one antiparasitic (Ivermectin) are analyzed. These molecules have been reported as possible drugs for the treatment of COVID-19. It is apparent that the molecular formulas do not have much in common. There are two molecules which have F atoms (Valrubicin and Eravacycline) and two with Cl atoms (Niclosamide and Amodiaquine). Tenofovir, Sofosbuvir and Remdesivir present P atoms in their formula. Ivermectin and Hesperidin are molecules formed from C, O and H and the other molecules also have N atoms in their structure. The action mechanism is different in the case of each drug, but apparently all are somewhat effective against COVID-19.

The electron donor–acceptor map (DAM) [38] was previously reported as a powerful tool for investigating antioxidant capacity via the electron transfer mechanism. The DAM is shown in Figure 2, together with the values of all the molecules being studied. In DAM, the good electron donor zone is down to the left and the good electron acceptor section is up to the right. All molecules can be classified according to DAM as either electron donors or electron acceptors. In this investigation, molecules with ω+ values above 1.0 are considered to be electron acceptors, and those with ω− values less than 5.5 are considered to be electron donors. Results from Figure 2 indicate that most of the molecules are electron donors rather than electron acceptors. As mentioned previously [40,41,42,43], electron donor and electron acceptor capacity are both important factors in the prevention of oxidative stress. Free radicals are very reactive molecules that present an unpaired electron and produce oxidative stress. Alternatively, electron acceptor molecules are free radical scavengers, which can accept an unpaired electron, thus deactivating the free radical. Acceptance and donation of electrons are both mechanisms that help scavenge free radicals; therefore, curtailing the oxidative stress that they produce. In this investigation, the best electron acceptors out of the molecules analyzed are Valrubicin (an anticancer drug with three F atoms), Eravacycline (an antibiotic with one F atom) and Niclosamide (an anthelmintic drug with two Cl atoms). Apparently, the presence of halogens increases electron-donating capacity and the number of halogen atoms is directly related to electron donor power. Moreover, it was discovered that the introduction of fluorine affects the replication of some viruses [25]. Therefore, these molecules with F atoms may be good antioxidants, while also helping curtail virus replication. Specifically, Valrubicin has been classified as an antitumor antibiotic, made from natural products produced by the soil fungus *Streptomyces.* This drug affects multiple phases during the cell cycle. The action mechanism may be related to its capacity to accept electrons. 

Amodiaquine is an antiviral drug with one Cl atom and is a good electron donor. Hesperidin has been reported as an antioxidant, concurring with the results presented in Figure 2, because it is located within the electron donor zone. Elbasvir is also a good electron donor, but its main action mechanism refers to its ability to inhibit the 5A protein. 

Ribavirin and Galidesivir are effective as potent drugs against COVID-19, as they bind tightly to its RdRp (a crucial viral enzyme in the life cycle of RNA viruses). They can also act as free radical scavengers because they donate electron. There are another three molecules (Remdesivir, Sofosbuvir and Tenofovir) that are good electron donors, but the reported action mechanism against COVID-19 is that they interact with RdRp. Carfilzomib is an antineoplastic agent and a selective proteasome inhibitor. It works by stopping or slowing the growth of cancer cells in the body and is also a good electron donor, but not as good as elbasvir. Notably, Carfilzomib and Valrubicin are both anticancer drugs, but the first is an electron acceptor while the second is an electron donor. Ivermectin and Molnupiravir are two potent drugs, as we discuss in the following, but they are not as good electron acceptors as valrubicin.

Theoretical SARS-CoV-2 protease binding energies of some of the drugs we are investigating have recently been reported [13,15,16]. Table 2 presents these values and also the electron donor and electron acceptor powers that we obtained. The idea is to correlate the electron donor–acceptor capacity with the binding energies. Apparently [16] whether drugs are efficacious against COVID-19 is related to their capacity to form bonds, as drugs that bind tightly to the protein better control virus replication. The first thing of note in Table 2 is that all binding energies are around 7 kcal, except for Carfilzomib, which presents 13.8 kcal/mol. This means that Carfilzomib binds strongly to the main SARS-CoV-2 protease. However, it is not the best electron acceptor but it is a good electron donor. Valrubicin is the best electron acceptor but its binding energy does not exceeds that of the other molecules. Galidesivir is a good electron donor, and its binding energy is similar to that of Tenofovir. The binding energy of Eravacycline is similar to the binding energy of Amodiaquine, but this molecule is neither the worst nor the best electron acceptor. This analysis indicates that there is no correlation between binding energy and the ability to accept or donate electrons. 

The Front Line COVID-19 Critical Care (FLCCC) Alliance [17] was created in March 2019 in response to the global health emergency, in order to review emerging basic science and the clinical quest for a COVID-19 treatment protocol. The FLCCC recently discovered the powerful antiviral and anti-inflammatory properties of Ivermectin, an antiparasitic drug. On 9 March 2021, the FLCCC alliance *applauds the recognition by an international group of medical experts of Ivermectin as a safe and effective treatment for COVID-19* [18,19]. In addition, more recently, MK-4482 (Molnupiravir), an anti-influenza virus, has been administered [26]. The apparent great effectiveness of Molnupiravir and Ivermectin against COVID-19 may indicate that control of the pandemic is imminent. Molnupiravir is a pro-drug that remedies the copying errors that are causing the virus catastrophe. The action mechanism of Ivermectin apparently relates to the competitive binding of this drug with the host receptor-binding region of the SARS-CoV-2 spike protein. Evidently, these two drugs do not function as antioxidants. Our results indicate that these are not good electron acceptors. Instead, both are electron donors, but Molnupiravir is worse than Ivermectin. 

Most antiviral drugs are electron donors, meaning they are molecules capable of reducing other molecules. As these molecules are capable of reducing other molecules, they can incapacitate and affect the conditions necessary for viral infection, such as the replication and spread of the virus, but likewise they may reduce other molecules essential for life. Thus, when choosing between these powerful drugs, one might assume that the multifunctional Ribavirin and Valrubicin are more effective than the others because they are better at trapping free radicals. However, valrubicin accepts electrons and may thus oxidize other molecules. The paradox is that the facility to oxidize other molecules may make it useful for preventing infection but also incur danger by oxidizing the molecules that are essential for life. Ivermectin and Molnupiravir; however, represent two electron donating drugs that are powerful against COVID-19, and although they are electron donors better than Ribavirin. This may therefore, be an advantage when considering these two molecules. All these drugs are somewhat effective for controlling the disease, but, up until now, there is no effective *weapon* against COVID-19.

## 4. Conclusions

Oxidative stress contributes to increasing the possibility of getting sick from COVID-19. Therefore, free radical scavengers may be useful for the treatment of COVID-19. To scavenge free radicals, a molecule can be either an electron donor or electron acceptor. According to the results reported here, the best electron acceptor is Valrubicin (an anticancer drug with three F atoms in the structure) and the best electron donor is elbasvir (antiviral for chronic hepatitis C). Most of the antiviral drugs are good electron donors. In order to choose between these powerful drugs, it is necessary to consider their multifunctionality and; therefore, assume that Ribavirin and Valrubicin are better than the others as they may also be better at trapping free radicals. However, valrubicin accepts electrons and; therefore, oxidizes other molecules. The paradox is that, by being able to oxidize other molecules, it may be useful for preventing infection but also represent a danger because it oxidizes the molecules essential for life. Ivermectin and Molnupiravir are two powerful drugs against COVID-19 that are electron donors better than Ribavirin. This may turn out to be an advantage in terms of these two molecules as drugs against COVID-19. All of these drugs are somewhat effective in controlling the disease, but, until now, no effective *weapon* against COVID-19 has been identified. Free radical scavengers can act as *bullets* against COVID-19 but may also act against the organism, a factor that should be taken into consideration. Looking ahead, clinical studies are needed to define the importance of antioxidants in treating COVID-19.

## Figures and Tables

**Figure 1 antioxidants-10-00979-f001:**
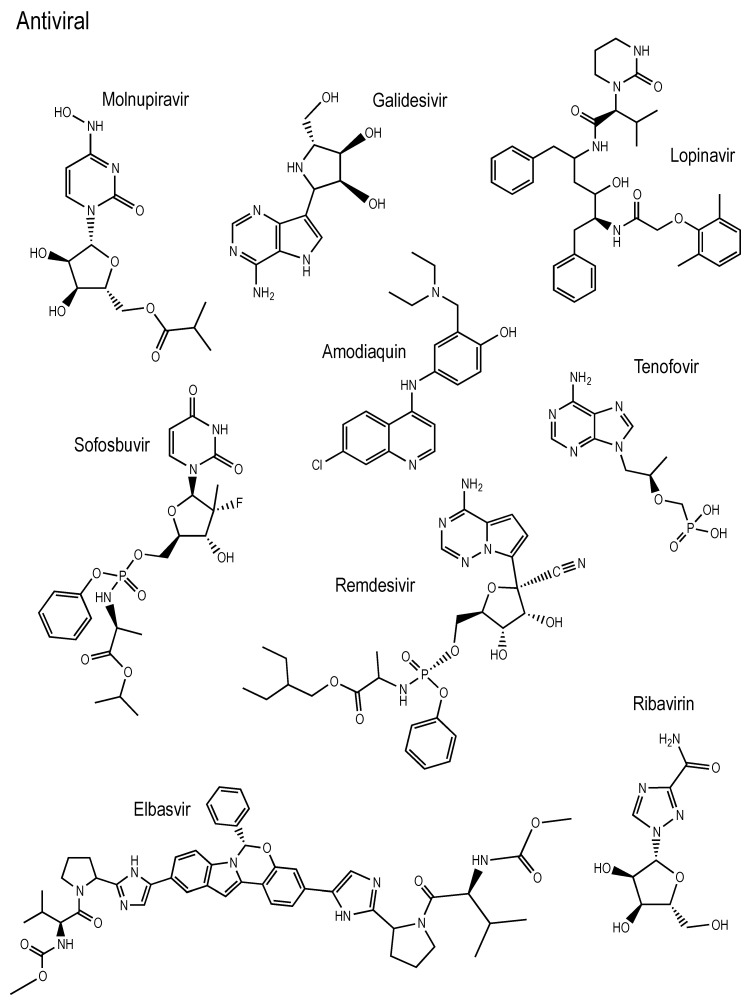
Schematic representation of the studied molecules.

**Figure 2 antioxidants-10-00979-f002:**
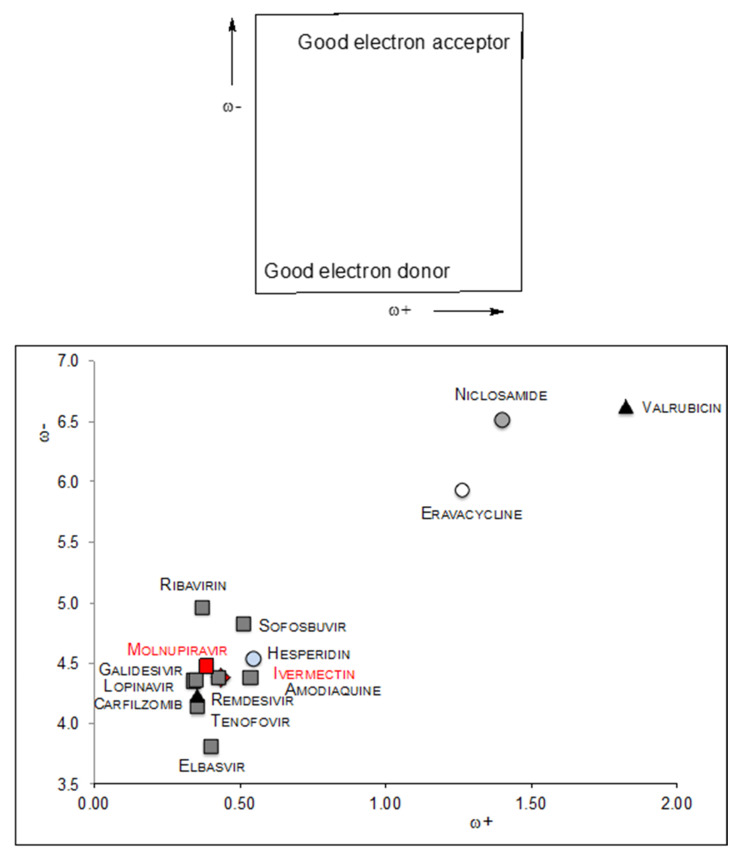
Donor-Acceptor MAP (DAM) of the studied molecules.

**Table 1 antioxidants-10-00979-t001:** Drugs used to control COVID-19.

Drug	Activity	Reference
CARFILZOMIB	Anticancer	[11]
VALRUBICIN	Chemotherapy	
ELBASVIR	Antiviral for chronic hepatitis C	[13]
HESPERIDIN	Anti-neurodegenerative, antioxidant, anti-neuro-inflammatory	[22]
NICLOSAMIDE	Anthelminthic	[3]
ERAVACYCLINE	Antibiotic	[13]
LOPINAVIR	Antiretroviral	[13]
RIBAVIRIN REMDESIVIR SOFOSBUVIR GALIDESIVIR TENOFOVIR	Antiviral for hepatitis C, Ebola, antiretroviral	[16]
AMODIAQUINE MOLNUPIRAVIR	Antiviral	[20]
IVERMECTIN	Antiviral, antiparasitic	[17,18]

**Table 2 antioxidants-10-00979-t002:** Electron donor and electron acceptor powers of selected molecules are reported. Red values are for electron acceptor molecules, whilst blue values are for electron donors. Absolute values of Binding Energies (BE in kcal/mol) previously reported are included (see [13,15,16]).

	−	+	BE
CARFILZOMIB	4.24	0.36	13.8
RIBAVIRIN	4.96	0.37	7.8
AMODIAQUINE	4.38	0.54	7.77
ERAVACYCLINE	5.92	1.26	7.7
REMDESIVIR	4.38	0.43	7.6
SOFOSBUVIR	4.83	0.51	7.5
VALRUBICIN	6.62	1.83	7.2
GALIDESIVIR	4.35	0.34	7.0
TENOFOVIR	4.14	0.36	6.9
LOPINAVIR	4.36	0.35	6.6
ELBASVIR	3.81	0.40	6.5

## Data Availability

Not applicable.

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
