# Peer review of "Electron Donor–Acceptor Capacity of Selected Pharmaceuticals against COVID-19"

_antioxidants, 2021, doi:10.3390/antiox10060979_

Round 1

Reviewer 1 Report

On my mind the basic hypothesis of the paper is incorrect because:

  • Association of COVID-19 with oxidative stress (OS), if exists, does not mean causal relationship, i.e. that OS can cause or affect COVID-19 (just as other pathological state). OS may be rather the sequence of pathology. In fact, no large intervention studies, excluding limited number of very special cases, have shown benefits of antioxidants supplementation.
  • The author claims that antioxidative activity is determined by donor/acceptor properties of molecule and that either potent oxidants or potent reductants are able to quench radicals and by this prevent/minimize oxidative OS. Really, most of antioxidants are reductants but I see no reason for strong deterministic relationship between oxidative/reductive properties and ability to prevent OS. In any case, reduction potentials can be measured experimentally without unevitable assumptions underlying computational methods.
  • In vivo enzymatic defence is the major factor preventing oxidative damage. Are concentration of drugs in circulation sufficient to protect targets of free attacks? I don’t think so.
  • Even if OS is minimized by some drugs, will this minimization beneficial with respect to health outcomes?

In conclusion, my recommendation is to reject the paper.

Author Response

Reviewer#1

On my mind the basic hypothesis of the paper is incorrect because:

  • Association of COVID-19 with oxidative stress (OS), if exists, does not mean causal relationship, i.e. that OS can cause or affect COVID-19 (just as other pathological state). OS may be rather the sequence of pathology. In fact, no large intervention studies, excluding limited number of very special cases, have shown benefits of antioxidants supplementation.

Author’s reply

There are reports relating OS with COVID-19 (see references 14, 22 and 24). There is a lot of information concerning COVID-19 and some information is contradictory including, as the reviewer said, the benefits of antioxidants supplementation. In any case, this paper reports a characterization of these drugs as electron donors or acceptors as it was already explained in the introduction with the following paragraph:

Thus, the main objective of this report is to evaluate the antioxidant properties of the molecules presented in Table 1. Single electron transfer mechanism is analyzed and a classification concerning the free radical scavenger capacity is provided. This information may help to identify the best weapons agains COVID-19.

Antioxidants are not a panacea. I do not claim to give THE SOLUTION on COVID-19 with this analysis, but this study presents valuable information that, along with everything else, contributes to the construction of knowledge.

  • The author claims that antioxidative activity is determined by donor/acceptor properties of molecule and that either potent oxidants or potent reductants are able to quench radicals and by this prevent/minimize oxidative OS. Really, most of antioxidants are reductants but I see no reason for strong deterministic relationship between oxidative/reductive properties and ability to prevent OS. In any case, reduction potentials can be measured experimentally without inevitable assumptions underlying computational methods.

Author’s reply

Reduction potentials can be measured but they can also be calculated accurately and with less expense. I agree that most antioxidants are reductants, but not all. Considering the electron acceptor capacity of substances give us another perspective that has been useful in other scenarios (references 40-47)

Reviewer#1

  • In vivo enzymatic defense is the major factor preventing oxidative damage. Are concentration of drugs in circulation sufficient to protect targets of free attacks? I don’t think so. Even if OS is minimized by some drugs, will this minimization beneficial with respect to health outcomes?

Author’s reply

These are very interesting questions but unfortunately, they are out of the scope of this investigation. To emphasize this idea we added the following paragraph at the end of abstract and conclusions

Looking ahead, clinical studies are needed to define the importance of antioxidants in treating COVID-19.

Reviewer 2 Report

The paper is interesting and well written. However, I suggest to discuss the importance of vitamin D in favoring immune response to infections as Covid19 and consequently to oxidative stress (see and add as reference papers by Murdaca et al concerning vitamin D and immune system and Covid19). Finally, it may be usefull in the discussion briefly discuss the role of vaccination against Covid19 also in patients with chronic ilnesses as autoimmune diseases to decrease the risk of morbility and mortality (see and add as reference paper by Ferri et al concerning italian rheumatic patients and Covid19).

Author Response

Reviewer#2

The paper is interesting and well written. However, I suggest to discuss the importance of vitamin D in favoring immune response to infections as Covid19 and consequently to oxidative stress (see and add as reference papers by Murdaca et al concerning vitamin D and immune system and Covid19). Finally, it may be usefull in the discussion briefly discuss the role of vaccination against Covid19 also in patients with chronic ilnesses as autoimmune diseases to decrease the risk of morbility and mortality (see and add as reference paper by Ferri et al concerning italian rheumatic patients and Covid19.

Author’s reply

This is an important recommendation. The following paragraphs and the references that were suggested by the reviewer are already included.

Effective vaccines to prevent infection have already been developed, also in patients with chronic illness as autoimmune diseases [2]

The data published in the literature on the effects of vitamin D supplementation are still controversial in patients with COVID-19, but the importance of vitamin D in promoting the immune response to infections such as Covid19 has been reported [4].

  1. Ferri, G.; Giuggioli, D.; Raimondo, V.; L’Andolina, M.; Tavoni, A.; Cecchetti, R.; Guiducci, S; Ursini, F; Caminiti, M.; Varcasia, G.; Gigliotti, P.; Pellegrini, R.; Olivo, D.; Colaci1, M.; Murdaca, G.; Brittelli, R.; Mariano, G.P.; Spinella, A.; Bellando-Randone, S.; Aiello, V.; Bilia, S.; Giannini, D.; Ferrari, T.; Caminiti, R.; Brusi, V.; Meliconi, R.; Fallahi, P.; Antonelli, A. COVID-19 and rheumatic autoimmune systemic diseases: report of a large Italian patients series. Clinical Rheumatology 2020, 39, 3195-3204.
  2. Murdaca, G.; Pioggia, G.; Negrini, S. Vitamin D and Covid-19: an update on evidence and potential therapeutic implications. Clin. Mol. Allergy 2020, 18:23, 1-8.

Reviewer 3 Report

The authors present a manuscript entitled “Potential pharmaceutical weapons consisting of antioxidant bullets against COVID-19”

The work is certainly interesting as it offers a chemical-physical approach for understanding the potential activity of antiviral drugs and for antiviral use, such as SARS-COV2. Reading the work, however, the critical comments of the authors do not seem very easily enucleated and, in my opinion, since there does not seem to be a strong correlation between the electron attractor-donor properties, the bond strength and the activity of the evaluated drugs, the authors they should better explain the usefulness and advantages of such an approach.

Minor points:

Please organize Tab. 1 better by presenting the action mechanism in another column and the relative reference in another. Develop the whole tab vertically

Pg 8: “free radial scavengers”, do you mean “free radical scavengers”?

Author Response

The authors present a manuscript entitled “Potential pharmaceutical weapons consisting of antioxidant bullets against COVID-19”

The work is certainly interesting as it offers a chemical-physical approach for understanding the potential activity of antiviral drugs and for antiviral use, such as SARS-COV2. Reading the work, however, the critical comments of the authors do not seem very easily enucleated and, in my opinion, since there does not seem to be a strong correlation between the electron attractor-donor properties, the bond strength and the activity of the evaluated drugs, the authors they should better explain the usefulness and advantages of such an approach.

Author’s reply

These results represent valuable information that, along with everything else, contributes to the construction of knowledge. The most important is to consider the electron acceptor capacity to prevent oxidative stress. This gives us another perspective that has been useful in other scenarios (references 40-47)

Table 1 is re-organized according to the suggestion.

Round 2

Reviewer 1 Report

I’ve noticed no significant changes in the revised version of the ms. My objections, as detailed in the review of the original version, related rather to the provocative title reflecting more than debatable approach. I suggest reconsidering the title to moderate formulation describing the actual content, e.g. “Electron donor-acceptor capacity of the selected pharmaceuticals as a potential factor of anti-COVID-19 activity”.

Author Response

The modified title of the manuscript is

Electron donor-acceptor capacity of selected pharmaceuticals against

COVID-19

Reviewer 3 Report

In my opinion, the manuscript in the present version is improved and clearer for readers. The critical points highlighted in the first round of review were well explained / corrected by the author.

Author Response

Thanks the reviewer for his/her comments.